# Hypothalamic melanin concentrating hormone neurons communicate the nutrient value of sugar

Ana I Domingos[1]*[†], Aylesse Sordillo[1], Marcelo O Dietrich[2,3], Zhong-Wu Liu[2], Luis A Tellez[4,5], Jake Vaynshteyn[6], Jozelia G Ferreira[4,5], Mats I Ekstrand[1], Tamas L Horvath[2], Ivan E de Araujo[4,5], Jeffrey M Friedman[1,7]*

[1]Laboratory of Molecular Genetics, The Rockefeller University, New York, United States; [2]Section of Comparative Medicine, Yale University, New Haven, United States; [3]Department of Biochemistry, Universidade Federal do Rio Grande do Sul (UFRGS), Porto Alegre, Brazil; [4]Feeding Laboratory, The JB Pierce Laboratory, New Haven, United States; [5]Department of Psychiatry, Yale University School of Medicine, New Haven, United States; [6]Department of Neuroscience, Albert Einstein College of Medicine, New York, United States; [7]Howard Hughes Medical Institute, The Rockefeller University, New York, United States

*For correspondence:
dominan@igc.gulbenkian.pt
(AID); friedj@rockefeller.edu
(JMF)

[†]Present address: Obesity
Laboratory, The Gulbenkian
Science Institute, Oeiras,
Portugal

Reviewing editor: Jeremy
Nathans, Howard Hughes
Medical Institute, Johns Hopkins
University School of Medicine,
United States

**Abstract** Sugars that contain glucose, such as sucrose, are generally preferred to artificial sweeteners owing to their post-ingestive rewarding effect, which elevates striatal dopamine (DA) release. While the post-ingestive rewarding effect, which artificial sweeteners do not have, signals the nutrient value of sugar and influences food preference, the neural circuitry that mediates the rewarding effect of glucose is unknown. In this study, we show that optogenetic activation of melanin-concentrating hormone (MCH) neurons during intake of the artificial sweetener sucralose increases striatal dopamine levels and inverts the normal preference for sucrose vs sucralose. Conversely, animals with ablation of MCH neurons no longer prefer sucrose to sucralose and show reduced striatal DA release upon sucrose ingestion. We further show that MCH neurons project to reward areas and are required for the post-ingestive rewarding effect of sucrose in sweet-blind *Trpm5*[−/−] mice. These studies identify an essential component of the neural pathways linking nutrient sensing and food reward.

## Introduction

Animals and humans generally prefer sugars containing glucose, such as sucrose, compared to non-nutritive sweeteners such as sucralose (*Smiciklas-Wright et al., 2002*; *Jacobson, 2005*; *Domingos et al., 2011*; *Sicher, 2011*) as a result of the post-ingestive rewarding effect of sucrose (*Domingos et al., 2011*). This post-ingestive rewarding effect of sucrose was first described by showing that non-nutritive liquids that are paired to glucose administration either in the intra-gastric tract or in plasma, are greatly preferred over liquids that are not paired with nutrients (*de Araujo et al., 2008*; *Ren et al., 2010*; *de Araujo et al., 2010*; *Oliveira-Maia et al., 2011*; *Sclafani et al., 2011*; *Fernstrom et al., 2012*; *de Araujo et al., 2013*). In addition, sweet-blind *Trpm5* knockout mice can still sense the nutrient value of sucrose (*de Araujo et al., 2008*). These studies have indicated that the nutrient value of sucrose is sensed and in turn establishes a preference for nutritive sugars (*Ren et al., 2010*; *de Araujo et al., 2010*; *Sclafani et al., 2011*; *Fernstrom et al., 2012*). These data further indicate that the post-ingestive rewarding effect plays an important role in driving nutrient choice (in addition to sweet taste). However, despite the substantial evidence that they play a major, perhaps dominant, role in driving food intake

**eLife digest** Sales of full-sugar fizzy drinks are almost triple those of diet versions, providing real-world confirmation of the laboratory finding that humans, as well as animals, prefer sugar to artificial sweeteners. However, it is not simply that sugary things taste better. Mice with a mutation that prevents them from perceiving sweet tastes still prefer the natural sugar sucrose over the artificial sweetener sucralose.

This is because sugar, unlike artificial sweeteners, has nutritional value, and both humans and animals find it rewarding to consume foods with a high caloric content. Consuming sugar has been known to cause certain parts of the brain to release more of the chemical transmitter dopamine, which is used to signal reward, but exactly how this process produces a preference for sugar has been unclear.

Now, Domingos et al. have revealed that a brain region called the lateral hypothalamus is responsible for this effect. This region of the brain—which helps to control appetite and which is also connected to the brain's reward system—normally contains cells called MCH neurons. Domingos et al. show that the natural preference for sucrose over sucralose can be reversed by stimulating the MCH neurons with light, which in turn stimulates dopamine release in reward centers in the brain. Moreover, mutant mice that do not have any MCH neurons in the lateral hypothalamus show a reduced preference for sucrose over sucralose, compared to normal mice, and they release less dopamine than normal mice when they consume sucrose.

By demonstrating that MCH neurons are both necessary and sufficient for sensing the nutritional value of sugar, these results provide new insights into the biological basis of sugar cravings. However, given the health implications of excessive sugar consumption, they may ultimately be used to find ways to make sugar less desirable, or to make artificial sweeteners more closely mimic the real thing.

and sweetener preference, the neural pathways that sense glucose and mediate the post-ingestive rewarding effect of sucrose have not been identified.

Rodent studies have further shown that sucrose but not artificial sweeteners such as sucralose can drive dopamine (DA) release in the midbrain even in the absence of taste (*de Araujo et al., 2008*; *Ren et al., 2010*; *de Araujo et al., 2010*; *Oliveira-Maia et al., 2011*; *Sclafani et al., 2011*; *Fernstrom et al., 2012*; *de Araujo et al., 2013*). The combination of sweet taste plus an increase of dopamine accounts for the preference for natural vs artificial sweeteners (*Domingos et al., 2011*). We previously reported that the artificial sweetener sucralose is preferred to sucrose only if supplemented by a proxy for this post-ingestive reward in the form of optogenetic activation of DA neurons (*Domingos et al., 2011*). However, the elements of the neural circuit that convey the post-ingestive rewarding effect of sucrose and activate DA neurons are unknown.

Melanin-concentrating hormone-expressing neurons (*Pmch* or, MCH neurons; in accordance with previous literature [*Shimada et al., 1998*; *Alon and Friedman, 2006*; *Kong et al., 2010*], we adopt the later nomenclature throughout this report) in the lateral hypothalamus (LH) are glucose sensitive, and show increased activity when extracellular glucose levels increase (*Burdakov et al., 2005*; *Kong et al., 2010*). *Pmch* knockout and MCH neuronal ablation lead to reduced body weight, indicating a critical role of these neurons in the regulation of energy balance (*Shimada et al., 1998*; *Whiddon and Palmiter, 2013*). In addition, MCH neurons send dense projections to reward centers in the striatum and midbrain where dopaminergic neurons are located (current report). This strong anatomical connection between MCH neurons and reward nuclei, as well as the fact that MCH neurons sense glucose levels, led us to hypothesize that these hypothalamic neurons could play a role in conveying the reward value of sucrose.

## Results

We first tested whether optogenetic activation of MCH neurons could alter an animal's preference for sucrose vs sucralose using a BAC transgenic *Pmch*-CRE mouse line that we generated (see 'Materials and methods'). *Pmch*-CRE mice were crossed to the channelrhodopsin-2 (ChR2) reporter mouse line B6;129S-*Gt(ROSA)26Sortm32(CAG-COP4*H134R/EYFP)Hze*/J (*Madisen et al., 2012*), herein abbreviated

Rosa26-LSL-ChR2-YFP, to generate *Pmch*-ChR2 mice. We characterized *Pmch*-CRE mice and thus confirmed tissue- and cell-specific expression of ChR2-YFP in MCH neurons as shown (*Figure 1A*). The YFP signal was seen in the LH with the characteristic appearance of MCH neurons (*Figure 1A*), and there was a 92 ± 8% overlap of YFP and MCH. In addition, 97 ± 3% of MCH neurons expressed ChR2-YFP (*Figure 1B*). Whole-cell patch-clamp recordings in slice preparations confirmed light-evoked spiking at 5, 10, and 20 Hz (*Figure 1C*), as well as during continuous light pulses of one second (*Figure 1D*). The spike rate of MCH neurons was higher with light pulses of 20 Hz vs 5 Hz. Note, glucose has been shown to evoke similar high-frequency bursting of MCH neurons (see inset, *Figure 1D*) (*Burdakov et al., 2005*). We also recorded voltage responses to consecutive pulses of continuous light in order to test the capacity of these cells to resist repeated trains of light stimulation (*Figure 1D*). Spike attenuation (*Figure 1D*, inset) during optogenetic stimulation was similar to what has been previously reported for glucose-triggered responses (*Burdakov et al., 2005*; *Kong et al., 2010*) in MCH neurons, and changes in membrane potential were resilient to optical stimulation (*Figure 1—figure supplement 1B*).

We implanted optical fibers into the LH of *Pmch*-ChR2 and *Pmch*-CRE control mice (respectively, ChR2[+] and ChR2[−]) and assayed their preferences in a series of two-bottle choice tests (*Figure 2*, *Figure 2—figure supplement 1* and 'Materials and methods'). After five licks at a designated sipper, laser pulses of 5 Hz, 20 Hz, were delivered for 1 s, followed by a refractory period of another second ('Materials and methods' and *Figure 2—figure supplement 1*). ChR2(+) and ChR2(−) mice had equal preference for water+laser vs water alone at all stimulation frequencies tested (*Figure 2A*). We next compared an animal's preference for sucrose vs sucralose plus optogenetic activation of MCH neurons (see 'Materials and methods' for the rationale of concentrations chosen). In the absence of ChR2 (gray bars in *Figure 2*) or at a low stimulation frequency of 5 Hz (*Figure 2A*), animals still preferred sucrose to sucralose. However, consistent with the greater effect of 20 Hz on spiking of MCH neurons in slice preparation, a light frequency of 20 Hz, as well as continuous light, inverted an animal's preference

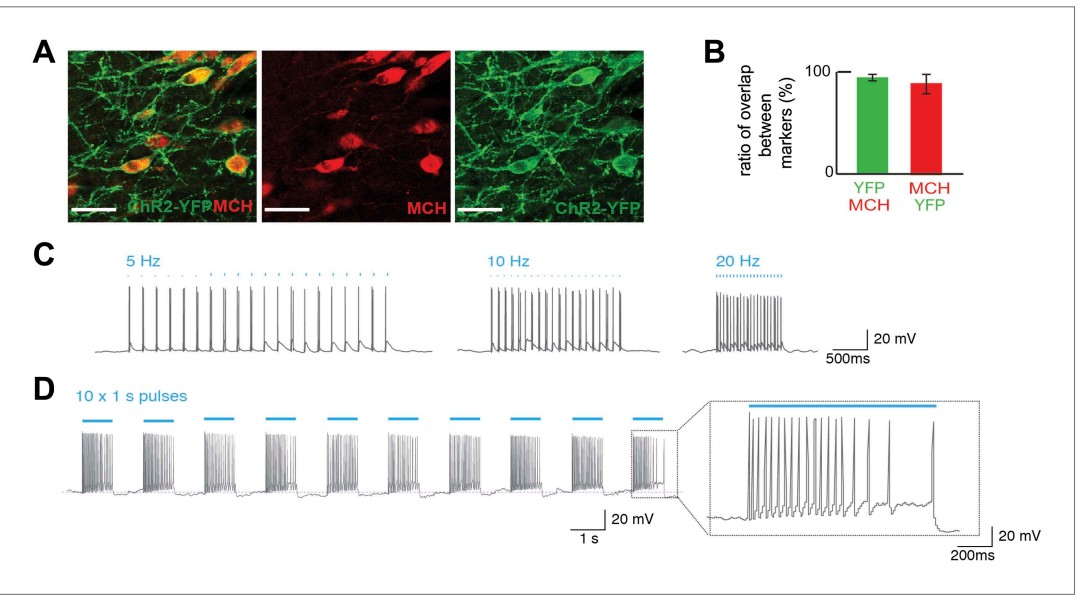

**Figure 1**. Optogenetic control of MCH neurons. (**A**) *Pmch*-CRE mice were mated to Rosa26-LSL-ChR2-YFP, and expression of ChR2-YFP (green—right panel) in MCH neurons (red—middle panel) in the LH are shown individually as well as in a merged panel (left panel); scale bar (Scale bar: 15 μm). (**B**) Quantification of co-expression of MCH and YFP shows that 97 ± 3% of MCH positive neurons expressed YFP and that 92 ± 8% of ChR2-YFP neurons expressed MCH (n = 1200 cells in four mice). (**C**) The effect of light stimulation on spike activity, evoked by light stimulation at 5 Hz, 10 Hz, and 20 Hz. (**D**) The response to 1 s continuous light stimulation, repeated 10 times. *Inset*, spike train in response to continuous light stimulation similar to what has been described for glucose-induced responses (see text for references and *Figure 1—figure supplement 1* for quantification).

The following figure supplements are available for figure 1:

**Figure supplement 1**. Optogenetic activation of MCH neurons.

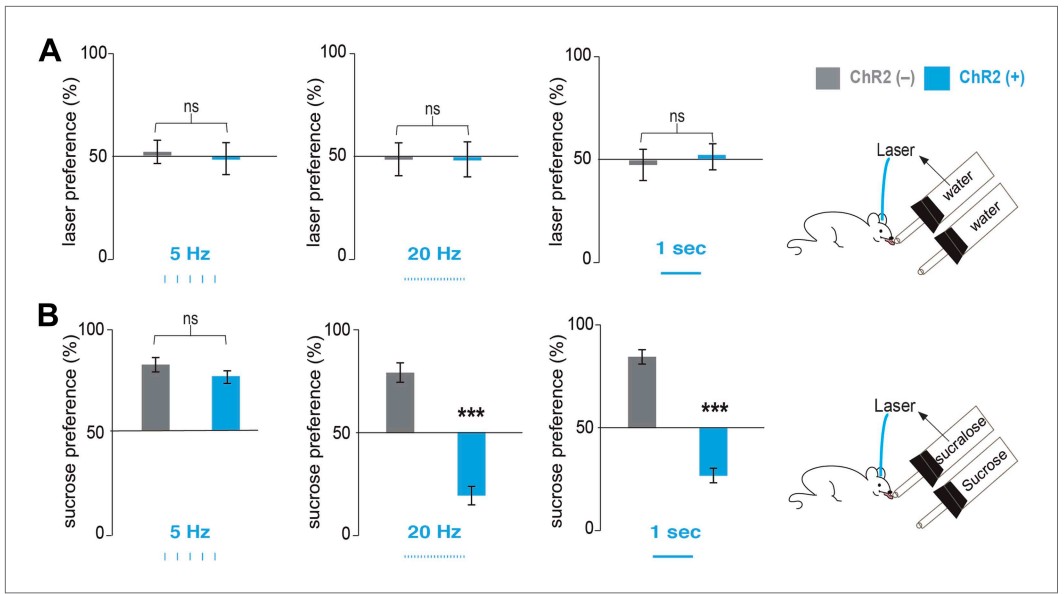

**Figure 2**. Optogenetic activation of MCH neurons inverts preference from sucrose to sucralose. (**A**) *Pmch*-ChR2 and *Pmch*-CRE control mice (respectively, ChR2[+] and ChR2[−]) were implanted with optical fibers (**Figure 2—figure supplement 1**) and were given the choice between water paired to laser and water alone. Laser preference is defined as the ratio of the number of licks of the water bottle that was paired to laser 'ON' and the total number of licks of both bottles (×100). Light stimulation during ingestive behavior was set to 5 Hz, 20 Hz, and continuous (**Figure 2—figure supplement 1** for lick/laser contingency). Optogenetic stimulation of MCH neurons during water intake did not influence preference behavior at any of the light stimulation frequencies. (**B**) ChR2(−) and Chr2(+) mice were given the choice between sucralose coupled to laser and sucrose. Sucrose preference is defined as the ratio of the number of licks of the bottle containing sucrose and the total number of licks of both bottles (×100). Light stimulation during ingestion of sucralose was set to 5 Hz, 20 Hz, and continuous. 20 Hz and continuous light stimulation, but not 5 Hz, inverts preference from sucrose to sucralose (see 'Materials and methods' and **Figure 2—figure supplement 2** for total licks per bottle). All data are mean ± SEM and n = 4 mice. ***p<0.0001, ns: p>0.28, *t* test.

The following figure supplements are available for figure 2:

**Figure supplement 1**. Lick/laser contingency.

**Figure supplement 2**. Total licks in **Figure 2**.

with a strong preference for sucralose plus MCH activation relative to sucrose (**Figure 2B**, **Figure 2—figure supplement 2**). At 20 Hz, ChR2(−) mice (**Figure 2B**, middle panel, gray bars) displayed a preference ratio for sucrose of 82.2 ± 3%, whereas ChR2(+) mice had a sucrose preference ratio of 20.0 ± 4% (**Figure 2B**, middle panel, blue bars). This preference ratio for sucrose is significantly lower than isopreference (p<0.0001 one sample T-test against 50%). Under continuous light, ChR2(−) mice displayed a preference ratio for sucrose of 76.7 ± 3% (**Figure 2B**, right panel, gray bars), whereas ChR2(+) mice displayed a preference ratio for sucrose of 26.8 ± 5% (**Figure 2B**, right panel, blue bars). This preference ratio for sucrose is significantly lower than isopreference (p<0.0007 one sample T-test against 50%).

The post-ingestive rewarding effect of glucose is associated with an increase of DA release in the striatum (**de Araujo et al., 2013**; **de Araujo et al., 2008**; **de Araujo et al., 2010**; **Ren et al., 2010**), and we confirmed that MCH neurons densely innervate the striatum and the ventral midbrain, making synapses onto DA neurons (**Figure 3—figure supplement 1**, 'Materials and methods'). We thus tested whether the inversion of preference to sucralose by activating MCH neurons was correlated with increases in striatal DA release as measured by microdialysis in ChR2(+) mice (**de Araujo et al., 2008**; **de Araujo et al., 2013**) (**Figure 3A**). After signal stabilization, samples were collected during a 30-min period when the animals had access to sucralose (**Figure 3A,B**). As previously reported, animals drinking

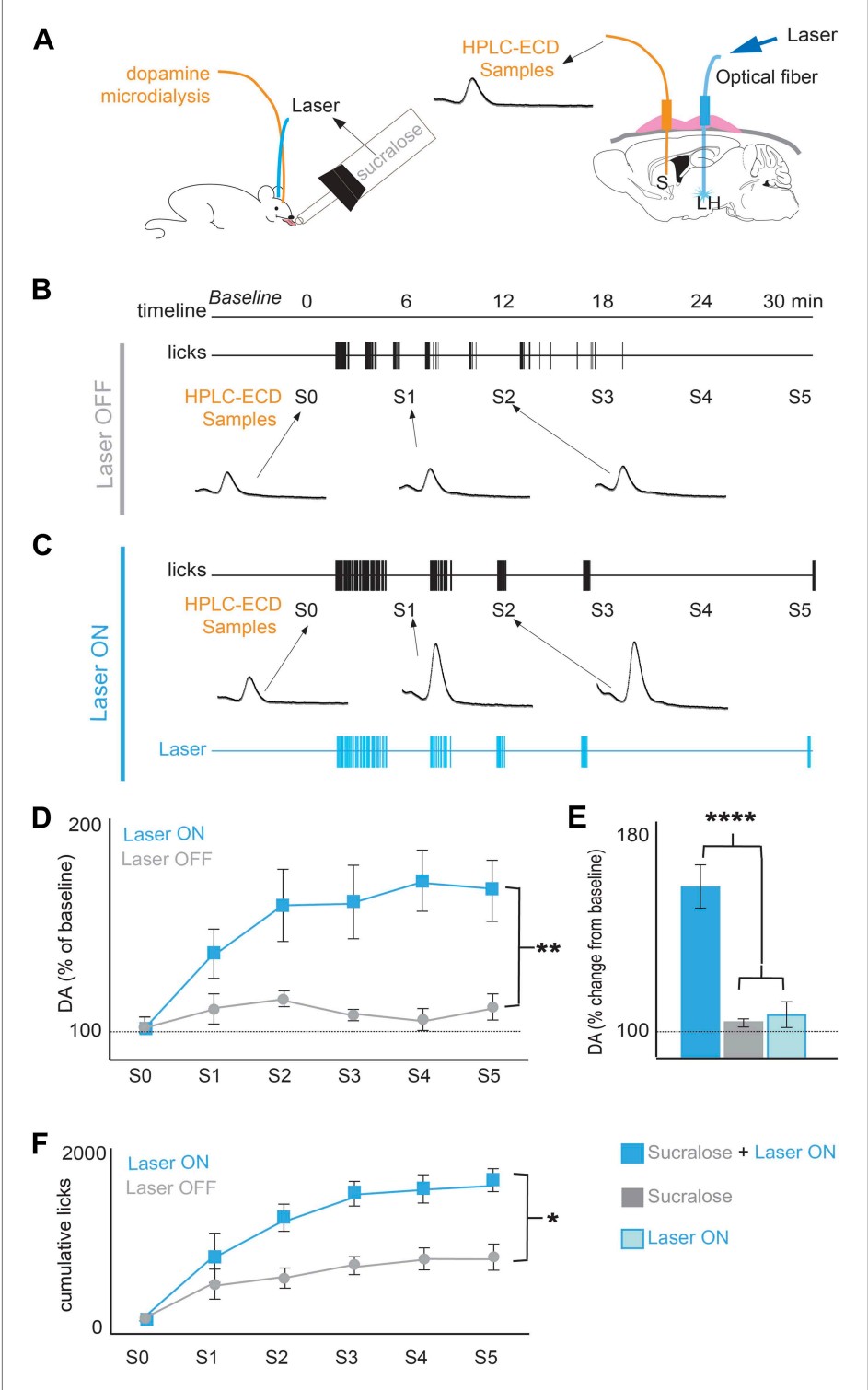

**Figure 3**. Optogenetic activation of MCH neurons increases DA release during sucralose ingestion. (**A**) Schematics of microdialysis sampling of striatal DA release in behaving mice (left panel) after intracranial implants of optical fibers in the LH and microdialysis probe in the striatum (S). (**B**) A timeline of licking behavior and DA collection with corresponding HPLC-ECD chromatograms of DA release when a ChR2(+) mouse drank 1.5 mM sucralose with the laser OFF. (**C**) A timeline with the laser ON at 20 Hz. (**D**) A timeline of average of DA increases from baseline across mice. (**E**) Overall change from baseline DA averaging across all S1–S5 samples in (**D**) and in the absence of drinking

*Figure 3. Continued on next page*

*Figure 3. Continued*

behavior with the laser ON (lighter blue). Each animal received the same number of pulses as the number of laser pulses delivered during ingestion of sucralose in the ON condition. On average, 201 ± 40 pulses were delivered. (**F**) The cumulative licks during microdialysis in both conditions are shown (see *Figure 3—figure supplement 1* for MCH projections to reward centers and *Figure 3—figure supplement 2* for requirement of DA transmission in sucrose/sucralose preference). All data are mean ± SEM and n = 4 mice, *p<0.05, **p<0.008, ****p<5.7e10$^{-7}$.

The following figure supplements are available for figure 3:

**Figure supplement 1**. MCH axonal projections onto DA neurons and reward areas.

**Figure supplement 2**. Preference for sucrose vs sucralose requires DA transmission.

sucralose (without light stimulation) displayed negligible changes in striatal DA levels (*Figure 3B*). Animals drinking sucralose when the laser was set to OFF displayed an overall 8.2 ± 2.6% change from baseline DA (average across all S1–S5 samples); this change was not significantly different from baseline (gray bars in *Figure 3E* p>0.05, one sample T-test against 100% baseline). However, when sucralose ingestion was coupled to laser stimulation of MCH neurons, DA release significantly increased in the striatum (**p<0.008; *Figure 3C–G*, blue bars). When laser was set to ON during ingestion of sucralose, DA levels increased 68.7 ± 9% vs baseline DA (average across all S1–S5 samples, blue bars in *Figure 3E*). This increase is not only significantly different from baseline and the OFF condition, but is also significantly different from DA release after experimenter-controlled (i.e., direct) delivery of laser pulses, in the absence of sucralose ingestion (light blue bar in *Figure 2E*, ****p<5.7e10$^{-7}$, t-test, with Bonferroni correction for multiple comparisons). Each animal received the same number of pulses as the number of laser pulses during ingestion of sucralose in the ON condition (*Figure 3E*). Mice used in condition laser-ON were the same as in condition laser-OFF. Optogenetic activation of MCH neurons markedly increased sucralose ingestion during the laser ON condition (*p<0.05; *Figure 3F*) while activation of MCH neurons did not change intake of water (*Figure 2—figure supplement 2*). As mentioned above, optogenetic stimulation of MCH neurons paired to water was not preferred to water alone, showing that activation of MCH neurons is not rewarding in the absence of sucralose. Thus, both MCH activation and the presence of sucralose were required to establish a change of preference.

We next tested whether a loss of MCH neurons decreased DA release during sucrose intake. We crossed *Pmch-CRE* mice to *C57BL/6-Gt(ROSA)26Sortm1(HBEGF)Awai*/J, (*Buch et al., 2005*) herein abbreviated Rosa26-LSL-DTR, to generate *Pmch*-CRE;LSL-DTR mice that specifically express the diphtheria toxin receptor (DTR) in MCH neurons (*Figure 4*). We injected *Pmch*-CRE;LSL-DTR mice with diphtheria toxin (+DT) or vehicle (+veh) intracranially (1 ng/g of body weight). This dose led to a complete loss of MCH neurons (*Figure 4A*, see *Figure 4—figure supplement 1* for other doses). MCH-ablated (*Pmch*-CRE-LSL-DTR+DT) and control mice (*Pmch*-CRE-LSL-DTR+veh, and LSL-DTR+DT) were subjected to microdialysis sampling of striatal DA during sucrose ingestion (*Figure 4B*). Control mice drinking sucrose displayed significantly higher dopamine levels compared to MCH-ablated mice (*Figure 4C–F*; **p<0.008 ANOVA in *Figure 4E*). Control mice showed an overall 118.3 ± 0.3% increase in striatal DA levels while drinking sucrose (average across all S1–5 samples, *Figure 4F*, ***p<1.98e10$^{-9}$ T-test). The increase of DA levels was significantly above those at baseline (p<0.0018, one sample T-test compared to 100% baseline). In contrast to animals without DT injection, MCH-ablated mice showed a negligible DA efflux during sucrose intake, and the levels after sucrose exposure did not differ from baseline levels (p>0.4, one sample T-test compared to 100% baseline). Baseline DA levels were similar in both groups (*Figure 4*, *Figure 4—figure supplement 2A*). Consistent with a lower reward value of sucrose, when given free access to sucrose or water, MCH-ablated mice consumed significantly less sucrose than control mice (*Figure 4G*, *p<0.05 ANOVA), while water intake was equivalent between the groups (*Figure 4*, *Figure 4—figure supplement 2B*).

MCH-ablated and control mice were also given a series of choices sequentially with studies of (**A**) sucrose vs sucralose, (**B**) sucrose vs water, (**C**) sucralose vs water (*Figure 5A–C*), and preference ratios for each comparison were computed. Preference ratios for sucrose in *Pmch*-CRE;LSL-DTR(+veh), LSL-DTR(+DT) and *Pmch*-CRE;LSL-DTR(+DT) mice were, respectively, 77.1 ± 7%, 82.0 ± 4%, and 39.9 ± 5%. While control mice preferred sucrose to sucralose, MCH-ablated mice no longer had a preference for

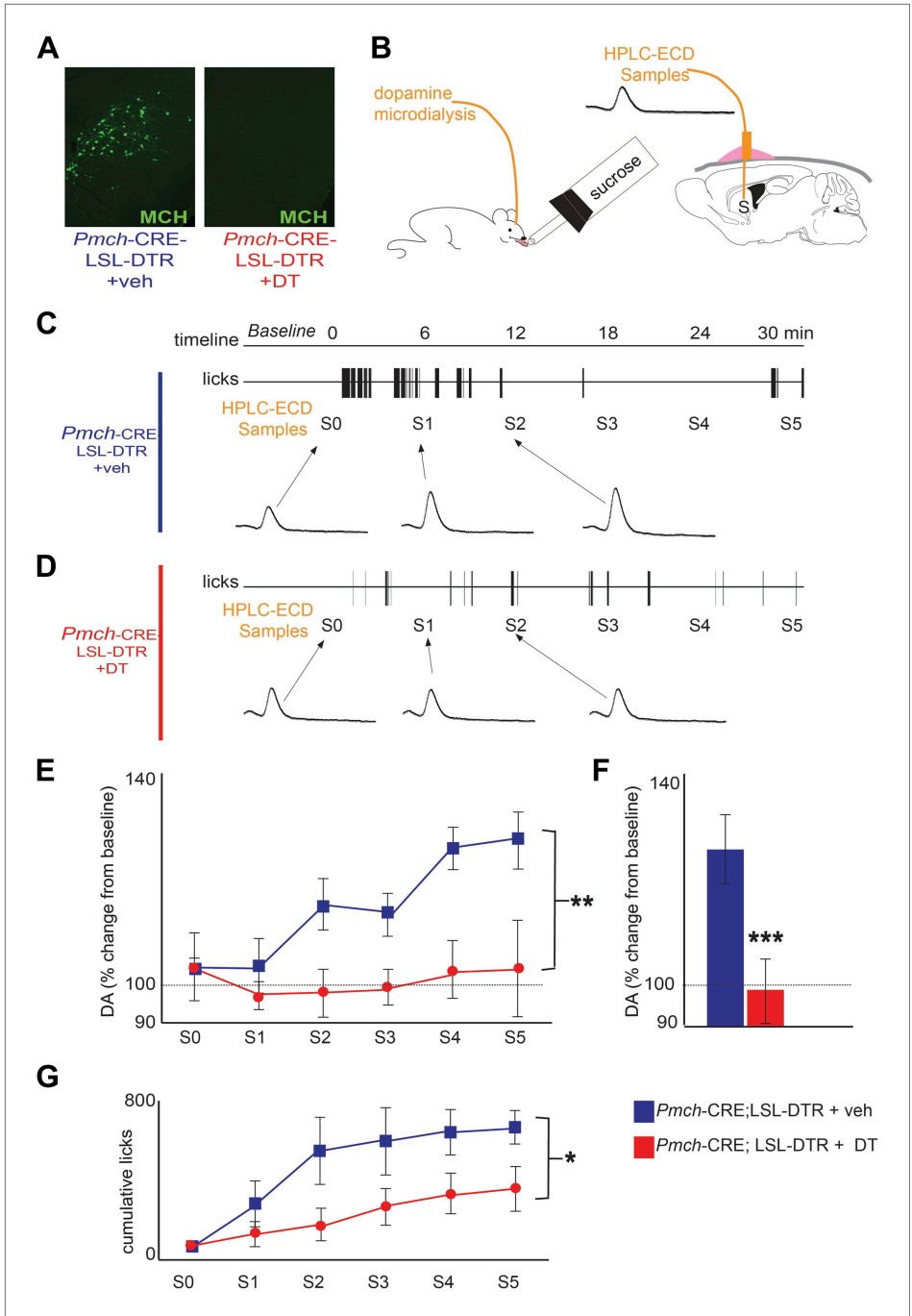

**Figure 4**. MCH neurons are required for DA release during sucrose ingestion. (**A**) *Pmch*-CRE;LSL-DTR mice were treated with 1 ng/g of DT. Complete ablation of MCH neurons by intracranial injection of diphtheria toxin is shown. (see *Figure 4—figure supplement 1* for other doses). (**B**) Schematics of microdialysis in behaving mice after intracranial implant of microdialysis probe in the striatum (S). (**C**) A timeline of licking behavior and DA collection, with corresponding HPLC-ECD chromatograms when a control mouse drank 0.4 M sucrose is shown. (**D**) A timeline similar to (**C**) when an MCH-ablated mouse drank sucrose (**E**) timeline of average DA increases from baseline across mice are shown. (**F**) For both genotypes, overall change from baseline DA averaging across all S1–S5 samples in (**E**) is shown. (**G**) Cumulative licks during microdialysis in both groups (see figure supplements for additional controls). All data are mean ± SEM and n = 4 mice. *p<0.05, **p<0.008, ***p<1.98e10$^{-9}$.

*Figure 4. Continued on next page*

*Figure 4. Continued*

The following figure supplements are available for figure 4:

**Figure supplement 1**. Titration of intracranial dose of DT.

**Figure supplement 2**. MCH-ablated mice have normal baseline DA levels and are not adipsic.

sucrose. This reduction in sucrose preference is statistically significant (*Figure 5A*, *p<0.0012, ¥p<0.009, T-test with Bonferroni correction for multiple comparisons, same-symbol pairs indicate statistically significant differences; see also *Figure 5—figure supplement 1* for total licks in each). However, MCH neuronal ablation did not alter an animal's preference for either sucrose or sucralose vs water indicating that, in contrast to their requirement for establishing the post-ingestive effect of sucrose, MCH neurons are not required for establishing a preference for sweet taste (*Figure 5B,C* and *Figure 5—figure supplement 1*). The postprandial increase in blood glucose after sucrose ingestion was normal in MCH-ablated animals (*Figure 5* and *Figure 5—figure supplement 2*), demonstrating that the changes in sucrose preference were not a result of any differences in blood glucose levels (*Kong et al., 2010*). Altogether, these data indicate that activation of MCH neurons are necessary and sufficient for establishing the preference of animals for sucrose compared to artificial sweeteners.

Finally, to confirm that MCH neurons are required for the post-ingestive rewarding effect of sugar even in the absence of sweet taste, we tested whether sucrose could condition preference in sweet-blind $Trpm5^{-/-}$ mice lacking MCH neurons (*Figure 5D*, *de Araujo et al., 2008*). Sweet-blind control mice showed a significant side bias towards the side where sucrose was placed during the conditioning sessions, in which sucrose or sucralose are delivered at opposite sides on alternate days, (*Figure 5D*). In contrast, sweet-blind MCH-ablated mice did not show a preference in this conditioning protocol, indicating that the post-ingestive rewarding effect of sucrose had been lost. $Trpm5^{-/-}$ *Pmch*-CRE;LSL-DTR+veh and $Trpm5^{-/-}$LSL-DTR+DT control mice showed conditioned side preferences of 70 ± 3% and 79 ± 5% respectively, towards the side where sucrose was placed. $Trpm5^{-/-}$ *Pmch*-CRE;-LSL-DTR(+DT) mice had 50 ± 7% conditioned side preference towards the side where sucrose was placed. This reduction in sucrose conditioning was statistically significant from that of controls (*Figure 5D*, *p<0.045, ¥p<0.099, T-test with Bonferroni correction for multiple comparisons, same-symbol pairs indicate statistically significant differences). Finally, the behavioral conditioning by the post-ingestive rewarding effect of sucrose in sweet-blind $Trpm5^{-/-}$ mice positively correlates with the extent of DA neuron activation, as assayed by staining for cFos in DA neurons in the ventral tegmental area (VTA) (*Figure 5*, *Figure 5—figure supplement 2*). These data confirm that MCH neurons are required for sensing the nutrient value of sucrose in the absence of taste.

## Discussion

In this manuscript, we report that MCH neurons are necessary and sufficient for establishing a preference for sucrose vs sucralose, an artificial sweetener. MCH neurons serve as an essential link between glucose sensing and sugar reward, and these data thus identify a key component of the neural circuit that establishes the preference for natural vs artificial sweeteners (*Smiciklas-Wright et al., 2002*; *Jacobson, 2005*; *Domingos et al., 2011*; *Sicher, 2011*).

MCH neurons have previously been shown to be excited by glucose (*Burdakov et al., 2005*; *Kong et al., 2010*), suggesting that direct glucose sensing by these neurons regulates reward. However, it is also possible that glucose is sensed elsewhere, such as by putative gastric/intestinal sensors or other nutrient sensors, which inform the brain and MCH neurons about the nutrient content of ingested food (*Sclafani et al., 2011*). Glucose sensing in MCH neurons or elsewhere would explain why, even in the absence of taste (such as in $Trpm5^{-/-}$ mice), sucrose is able to drive significant increase of striatal DA. This increase, in turn, conveys reward and conditions behavior (*de Araujo et al., 2008*). We note, however, that optogenetic stimulation of MCH neurons alone is not sufficient to alter behavior in the absence of taste. The data thus suggest that MCH neurons are components of a reward-encoding network that integrates information from multiple sources, including the nutrients themselves, lingual taste buds and, possibly, other sites of glucose sensing like the gut. Consistent with this possibility, viral tracing from lingual taste buds shows that MCH neurons are part of a circuit processing gustatory information (*Pérez et al., 2011*). This finding is also consistent with the optogenetic data reported

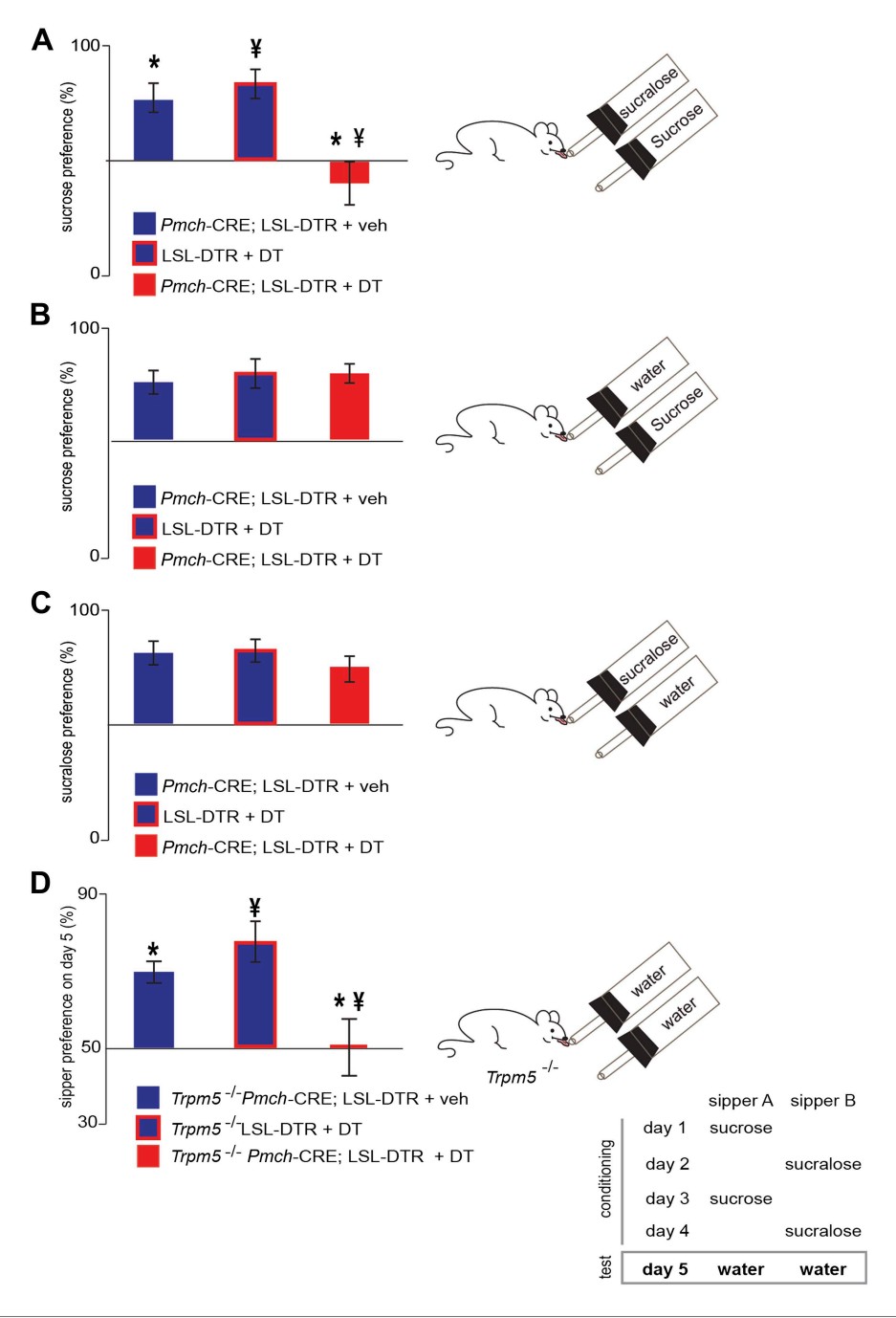

**Figure 5**. MCH neurons are required for sucrose vs sucralose preference, even in the absence of taste. (**A**–**C**) Mice with ablated MCH neurons (red filled bars) and their respective controls (blue filled bars) were given the choice of (**A**) 0.4 M sucrose vs 1.5 mM sucralose (*p<0.03, ¥p<0.011, see 'Materials and methods' for the rationale of concentrations). (**B**) Sucrose vs water. (**C**) Sucralose vs water. All mice preferred either sweetener—sucrose or sucralose—over water. (**D**) Sweet-blind $Trpm5^{-/-}$ mice, with and without ablation of MCH neurons, were subject to a 4-day bottle-conditioning protocol, in which sucrose and sucralose were presented in opposing bottles, on alternate days. Bottle preference was tested on the fifth day with two bottles filled with water. Sweet-blind control mice showed a significant side bias towards the bottle where sucrose was placed during the conditioning sessions whereas MCH-ablated mice did not (*p<0.045, ¥p<0.099, see **Figure 5—figure supplement 1** for total licks in each bottle, **Figure 5—figure supplement 2** for blood glucose controls, and 'Materials and Methods' for details). All data are mean ± SEM and n = 8 mice, $t$ test with Bonferroni correction for multiple comparisons.

*Figure 5. Continued on next page*

*Figure 5. Continued*

The following figure supplements are available for figure 5:

**Figure supplement 1**. Total licks per bottle in MCH-ablated and control mice.

**Figure supplement 2**. Ablation of MCH neurons affects neither peak blood glucose after IP challenge, nor acute postprandial blood glucose.

**Figure supplement 3**. Ablation of MCH neurons affects postingestive DA neuron activation.

here, showing a requirement for both sweet taste and activation of MCH neurons to drive reward. The role of MCH neurons uncovered here contrasts with that of DA neurons, which upon optogenetic stimulation have been shown to be rewarding when paired to water (*Domingos et al., 2011*).

A synergy between taste and the post-ingestive rewarding effect would explain why sucrose and other fructose/glucose disaccharides, which are more potent stimulators of sweet-taste receptors than glucose alone, are generally preferred to glucose alone (*Sclafani and Mann, 1987*; *Nelson et al., 2001*). This synergy would also explain why our optogenetic gain of function experiments lead to an inversion of preference, rather than an isopreference, which could perhaps have been achieved by decreasing the concentration of sucralose. Likewise, the loss of function of MCH neurons leads to isopreference, but this could perhaps be biased toward sucralose by increasing its concentration. Further studies will be necessary to establish the relevant sites of glucose sensing, identify additional elements of the neural circuit integrating gustatory perception with nutrient sensing and reward, as well as elucidating the neural mechanisms by which MCH neurons regulate striatal DA release. Several methods including viral tracing can be used to identify monosynaptic or polysynaptic inputs onto MCH and DA neurons.

As mentioned, the activity of MCH neurons is increased by glucose (*Burdakov et al., 2005*). Glucose-activated MCH neurons and pancreatic β cells share signal transduction components necessary for glucose sensing, and both Kir6.2 and UCP2 regulate glucose excitability of MCH neurons (*Kong et al., 2010*). Moreover, a loss of function of Kir6.1 in MCH neurons leads to alterations in results of a glucose tolerance test with increased plasma glucose at later times, establishing a role for these neurons in glucose homeostasis (*Kong et al., 2010*). Further studies are required to establish whether these or other components of glucose sensing pathways are also required for the ability of MCH neurons to influence sucrose preference. It is possible, however, that additional neural populations are components of this nutrient sensing circuit. For example, orexin/hypocretin-containing neurons can also sense glucose and it is thus possible that these or other neural populations in the mesolimbic system, or higher order centers can also influence the reward value of sugar (*Burdakov et al., 2005*; *Karnani and Burdakov, 2011*).

Previous reports have also explored the relationship between MCH neurons and reward: both MCH knockout and MCH ablated mice show augmented locomotor responses to psychostimulant drugs (*Pissios et al., 2008*; *Whiddon and Palmiter, 2013*). These augmented locomotor phenotypes contrast with the behavioral effect we see with a loss of sucrose preference in MCH ablated mice. It is possible that the locomotor phenotypes in response to stimulants result from actions in the ventral striatum, where the MCH receptor (MCHR-1) is expressed, and that sucrose preference relies on other brain areas. Further experiments will be required to ascertain which brain areas and MCH projections are relevant for sucrose preference. Loss of function of MCHR-1 recreates the locomotor phenotypes seen in *Pmch*[−/−] mice: *Mchr1*[−/−] mice are super-sensitive to the locomotor activating effects of d-amphetamine (*Smith et al., 2005*). These studies do not establish whether it is MCH or another neurotransmitter expressed in these neurons that is responsible for the observed phenotypes, and further experiments will also be required to ascertain whether the MCH neuropeptide itself is relevant for sucrose preference.

We assayed dopamine release using microdialysis to show that MCH neural activation increases dopamine release in the striatum and that the increase of dopamine in response to sucrose is lost after ablation of MCH neurons. Consistent with this, the levels of cFos in dopaminergic neurons of the VTA are reduced in MCH-ablated, sweet-blind *Trpm5* KO mice given sucrose. These assays of taste-blind mice confirm that MCH neurons regulate the activity of dopaminergic neurons, though the data do not establish whether this effect of dopaminergic neural activity and dopamine release is direct and/or

indirect. Further studies will be necessary to establish how MCH neurons regulate dopaminergic signaling. The delineation of this neural circuit may also provide a basis for understanding how leptin modulates reward (*Domingos et al., 2011*). The effects of leptin on reward are unlikely to be a result of a direct effect on MCH neurons as they do not appear to express the leptin receptor (*Leinninger et al., 2011*). However, a distinct neural population in the LH expressing neurotensin respond to leptin, and further studies may reveal whether or not leptin reduces MCH activity indirectly by activating these cells (*Leinninger et al., 2011*). Ablation of MCH neurons attenuates the obese phenotype of leptin deficient ob/ob mice indicating that MCH neurons are downstream of leptin action (*Alon and Friedman, 2006*).

Ablation of MCH neurons causes hypophagia and leanness, and it is possible that the reduced food intake is a result of a loss of the reward value of nutrient in these animals (*Alon and Friedman, 2006*). The reward value of sugar is also regulated by leptin, which has been recently reported to have a presynaptic action to suppress excitatory synaptic input onto VTA DA neurons (*Domingos et al., 2011*; *Thompson and Borgland, 2013*). Further studies will reveal whether leptin modulates excitatory output to the VTA via MCH neurons or influences nutrient preference by a direct effect on DA neurons in the VTA. The importance of brain nutrient sensing for behavior has also been studied in Drosophila (*Dus et al., 2011*, *2013*; *Miyamoto et al., 2012*, *2013*). Future studies will likely elucidate the extent to which the cellular mechanisms and neural pathways that regulate nutrient preference are shared between invertebrates and mammals. Brain nutrient sensing may represent an evolutionary adaptation to avoid starvation, by expediting decisions about which foods to consume.

In summary, these results confirm that MCH neurons are both necessary and sufficient for sensing the nutrient value of sucrose and suggest that these neurons play a critical role in establishing nutrient preference. The market share of sugared soda is nearly triple that of diet soda (*Smiciklas-Wright et al., 2002*; *Jacobson, 2005*; *Sicher, 2011*), and our data suggest a biological approach to potentially regulate sugar consumption. This could be achieved via the development of means for suppressing the activity MCH neurons, or via the development of new artificial sweeteners with neuroexcitatory activity specific to MCH neurons.

## Materials and methods

### *Pmch*-CRE transgenic mice

In order to restrict Cre expression to MCH neurons we used a BAC clone containing the full-length pro-melanin-concentrating hormone gene (RP23–129A21) with upstream and downstream flanking sequences of 108 kb and 89 kb, respectively. Prior to further manipulation, BAC DNA was prepared and electroporated into *E.coli* strain SW102 as required for BAC recombineering. An NLS-Cre PolyA construct (pML78, Mark Lewandowski, National Cancer Institute) was targeted to replace the ATG translational start codon of MCH exon 1 and correct insertion was verified by PCR and sequencing. 5′ recombineering homology: TGAAAGTTTTCATCCAATGCACTCTTGTTTGGCTTTATGCAAGCATCAAA 3′ recombineering homology: CTGCAGAAAGATCCGTTGTCGCCCCTTCTCTGGAACAATACAAAAA CGAC. All DNA fragments used for recombineering were generated with the FastStart High Fidelity PCR System (Roche, Indianapolis, IN). The modified BAC insert was released by NotI digestion, gel purified and used for pronuclear injection. Rosa26-LSL-ChR2-YFP and Rosa26-LSL-DTR were obtained from Jackson Laboratories. All animal procedures were carried out in accordance with the National Institutes of Health Guidelines on the Care and Use of Animals and approved by the Rockefeller University Institutional Animal Care and Use Committee (Protocols #13608, #10005 and #09012), JB Pierce Institutional Animal Care and Use Committee, (Protocol #101) and Yale University Institutional Animal Care and Use Committee (Protocol #2011-07942).

### Immunohistochemistry

Immunohistochemistry was performed as published elsewhere (*Domingos et al., 2011*, *Pérez et al., 2011*), using chicken anti-GFP (1:1000; Abcam, Cambridge, MA), rabbit anti-MCH (1:1000; Abcam), c-Fos (1:100; Abcam).

### Electrophysiology

30 to 50-day old MCH-ChR2-YFP mice were used for recordings. Mice were euthanized at the beginning of the light cycle (9:00 AM), and brain slices containing the LH were cut at 300 µm (2/mouse). Slices

were transferred to a chamber at room temperature to stabilize in artificial cerebrospinal fluid (aCSF). Slices were then transferred to a recording chamber after ≥1 hr recovery and constantly perfused at 34°C with bath solution at a speed of 1.5 ml/min. Whole cell patch-clamp recording was performed on identified MCH-YFP neurons with a Multiclamp 700B amplifier (Axon Instruments, New York, NY). The patch pipettes were made of borosilicate glass (Sutter Instruments, Novato, CA) with a Sutter pipette puller (P-97). The tip resistance of the recording pipettes was 2–3 MΩ after being filled with a pipette solution containing (in mM): K-gluconate 125, $MgCl_2$ 2, HEPES 10, EGTA 0.2, Mg-ATP 4, Na2- phosphocreatin 10, and Na2-GTP 0.5, pH 7.3 with KOH. The composition of the bath solution was as follows (in mM): NaCl 124, KCl 3, $CaCl_2$ 2, $MgCl_2$ 2, $NaH_2PO_4$ 1.23, glucose 2.5, sucrose 7.5, and $NaHCO_3$ 26. After a gigaohm seal and whole-cell access were achieved, membrane potential and action potentials were recorded under current clamp at 0 pA. ChR2 currents were recorded under voltage clamp mode. Light stimulation (470 nm, LED)(CoolLED pE-100, UK) was performed in the following configurations: 5, 10, 20 Hz (1 ms pulses, 20 pulses total); 10 x 1 s (1 s pulse light ON, 1 s light OFF). All data were sampled at 3–10 kHz and filtered at 1–3 kHz with an Apple Macintosh computer using Axograph X (Axograph X, Berkeley, CA).

## Behavioral and optogenetic setup

MedAssociates chambers (MedAssociates, St. Albans, VT) were equipped with two contact lickometers and a laser source (solid state Crystal laser, 473 nm wavelength) controlled by MedPC via a TT impulse to be triggered upon lick detection (*Domingos et al., 2011*). The laser turns on every five consecutive licks on the same bottle (*Figure 1—figure supplement 1*).

Animals were acclimated to the chambers until side preference for either bottle was even. During the acclimation and exposure periods mice were water deprived for 16–23 hr and were given water through the bottles inside the chamber for half an hour. In addition, stimuli in 10 min two-bottle tests were side balanced across the same genoptype group, being received either through the left bottle or through the right bottle. Two-bottle preference was calculated as the ratio: preference for 1 = number of licks on bottle 1/(number of licks on bottle 1+number of licks on bottle 2) and expressed as percentage values, with 50% representing the indifference ratio (referred to as isopreference in the 'Results' section). Behavioral data was analyzed with Excel and Prism, and expressed as mean ± SEM. Significance tests comparing groups were ANOVAs or *t* tests and, when appropriate, followed by Bonferroni corrections for multiple comparisons. Two-bottle tests without laser stimulation were carried out in the same setup, with the laser turned off. The size of each animal group is represented by 'n', and each animal was tested three times. The investigator was blind to the genotype. In all cases, concentration of sucrose was 0.4M and concentration of sucralose was 1.5 mM. These concentrations were based on previous literature (consult supplementary figure-4 in *Domingos et al., 2011*). Briefly, the differences in molarity of sucrose and sucralose reflect differences in ligand-binding affinity of either sweetener to taste receptors, and were chosen among the plateau values of behavioral dose-response curves (preference for either sweetener versus water in *Domingos et al., 2011*). Volume dispensed by the lickometers averages 2 μl/lick (*Domingos et al., 2011*). Locations of optical probes were confirmed histologically (data not shown). For each light stimulation regimen in *Figure 2*, mice in top and bottom panels are the same. After animals were corrected for any spontaneous side bias, and prior to the 10-min testing data in *Figure 2*, animals had a 10-min pre-exposure to either one of the two stimuli in two consecutive days. The pre-exposure procedure is intended to avoid novelty-related artifacts. On day one animals had exposure to water, followed by water+laser the day after. On the third day animals were tested for water vs water+laser for 10 min. On the fourth day, animals had exposure to sucrose, followed by sucralose+laser the day after. On the sixth day animals were tested for sucrose vs sucralose+laser for 10 min.

## Microdialysis during ingestive behavior

During the experimental sessions microdialysate samples from the freely-moving mice were collected, separated and quantified by high-pressure liquid chromatography coupled to electro-chemical detection methods ('HPLC-ECD'). Briefly, after recovery from surgery and behavioral habituation, a microdialysis probe (2 mm CMA-7, cut off 6 kDa, CMA Microdialysis, Stockholm, Sweden) was inserted into the striatum through the guide cannula (the corresponding CMA-7 model). After insertion, probes were connected to a syringe pump and perfused at 1.2 μl/min with artificial CSF (Harvard Apparatus). After a 90 min washout period, dialysate samples were collected every 6 min and immediately

manually injected into a HTEC-500 HPLC unit (Eicom, Japan). Analytes were then separated via an affinity column (PP-ODS, Eicom), and compounds subjected to redox reactions within an electro-chemical detection unit (amperometric DC mode, applied potential range from 0 to ~2000 mV, 1 mV steps). Resulting chromatograms were analyzed using the software EPC-300 (Eicom, Japan), and actual sample concentrations were computed based on peak areas obtained from 0.5 pg/µl dopamine stand-ards (Sigma) and expressed as % changes with respect to the mean dopamine concentration associ-ated with baseline (i.e., behavioral task) samples. Animals were water deprived for 16–23 hr, and rested in their home cages for baseline sample collection until values were stable. Chromatograms shown in *Figures 3 and 4* are time-gated to the DA peak at 1.7 min. S0 denotes the pre-ingestion sample, and refers to the sample in which the animal was placed inside the behavioral box. Locations of microdialysis probes were confirmed histologically.

### Electron microscopy of MCH synapses onto DA neurons

The pre-embedding dual-labeling protocol of anti-GFP and anti-TH used in this study was adapted from *Lane et al. (2010)*. Briefly, vibratome sections were placed in 0.1% sodium borohydride and 0.1% glycine in 0.1 M phosphate buffer to remove excess aldehydes. Sections were incubated in a cryopro-tectant solution (25% sucrose and 2.5% glycol in 0.05 M phosphate buffer), then immersed succes-sively in liquid Freon and liquid nitrogen to freeze, and thawed at room temperature in 0.1 M phosphate buffer to enhance penetration of immunoreagents. Sections were incubated in 2% bovine serum albumin (BSA) in PBS to block non-specific labeling and then incubated for 42 hr at 4°C in a primary antibody solution containing both rabbit anti-TH (P40101; 1:1000; Pel-Freez) and mouse anti-GFP (1:1000; Millipore mab3580) antibody in 0.1% BSA in PBS. Detection of GFP was done first. Sections were incubated for 2 hr in biotinylated horse anti-mouse IgG (1:1000) and the immunoperoxidase–DAB procedure was applied using avidin-biotin complex (Vectastain Elite ABC kit from Vector Laboratories), followed by diaminobenzidine and urea tablets (Sigma) for 10 min. The DAB reaction product was then silver-gold enhanced for 15 min using the Teclemariam method (*Teclemariam-Mesbah et al., 1997*). After fixation in 0.5% glutaraldehyde, the detection for TH began: the sections were incubated for 2 hr in biotinylated horse anti-rabbit IgG (1:1000) and followed by the same steps used for GFP except, no silver enhancement was used. After post-fixation in 1% osmium tetroxide/1% Potassium ferrocyanide in 0.1 M cacodylate buffer (pH 7.4) for 1 hr at 4°C, the sections were dehydrated in a graded ethanol series, propylene oxide and embedded in Eponate (Ted Pella, INC). Blocks were cut with a diamond knife on a Leica UltracutE. Ultra-thin (~70 nm) sections were collected on uncoated 200 mesh grids. Unstained sections were viewed with a TecnaiSpiritBT Transmission Electron Microscope (FEI) at 80 KV and images were taken with Gatan 895 ULTRASCAN Digital Camera.

### Blood glucose measurements

Blood glucose tests were performed on mice that had been fasted for 24 hr beginning at the onset of the dark cycle. The following day mice were given an intraperitoneal injection of an aqueous solution of 10% glucose (10 ml/Kg body weight) and blood glucose was measured from the tail vein at 0 and 10 min using an Ascensia Elite XL glucometer (Bayer Health-Care, Tarrytown, NY).

### Conditioning to post-ingestive rewarding effects of sucrose, in the absence of taste

Once acclimated to the behavioral chamber, sweet blind *Trpm5*[−/−] mice were resented with one bottle containing sucrose or sucralose. The drinking behavior is quantified by monitoring licks with contact lickometers (MedAssociates) and the number of licks for each bottle was used to calculate the preference ratio for sucrose, as in *Domingos et al. (2011)*. To verify whether *Trpm5*[−/−] mice with dysfunctional MCH neurons could detect the post-ingestive effects of sucrose, we adapted a conditioning protocol as in *de Araujo et al. (2008)*, that allows the animal to manifest taste-independent preferences (scheme in *Figure 5*). All experiments were conducted with naive animals under a 16–23 hr water deprivation schedule. Animals were conditioned for 4 days with daily 30 min sessions of free access to either 1.5 mM sucralose or 0.4 M sucrose in one-bottle forced-choice training sessions. Either solution was presented on the opposite sides of the chamber on intercalated days. After training, on the 5th day, side bias was tested in 10-min two-bottle water versus water tests. This procedure was executed in *Trpm5*[−/−] mice with ablated MCH neurons and control *Trpm5*[−/−] mice with normal MCH neurons.

## Acknowledgements

We thank Charles Zuker for providing the *Trpm5*$^{-/-}$ mouse line. We thank the JPB Foundation, The Klarman Family Foundation for Eating Disorders, The Rockefeller Foundation, CNPq (Brazil), and NIH for supporting this research.

## Additional information

### Funding

| Funder | Grant reference number | Author |
|---|---|---|
| JPB Foundation | | Jeffrey M Friedman |
| Klarman Family Foundation for Eating Disorders | | Ana I Domingos, Aylesse Sordillo, Jake Vaynshteyn, Jeffrey M Friedman |
| The Rockefeller Foundation | | Ana I Domingos, Mats I Ekstrand, Jeffrey M Friedman |
| Science and Technology Foundation- Portugal | | Ana I Domingos |
| CNPq–Brazil | | Marcelo O Dietrich |
| National Institutes of Health | DC009997, DK085579 | Ivan E de Araujo |
| National Institutes of Health | DP1 DK006850 | Tamas L Horvath |

The funders had no role in study design, data collection and interpretation, or the decision to submit the work for publication.

### Author contributions

AID, Conception and design, Acquisition of data, Analysis and interpretation of data, Drafting or revising the article, Contributed unpublished essential data or reagents; AS, MOD, Z-WL, LAT, JV, JGF, Acquisition of data, Analysis and interpretation of data, Drafting or revising the article; MIE, Analysis and interpretation of data, Drafting or revising the article, Contributed unpublished essential data or reagents; TLH, IEA, JMF, Conception and design, Analysis and interpretation of data, Drafting or revising the article

### Ethics

Animal experimentation: All procedures were carried out in accordance with the National Institutes of Health Guidelines on the Care and Use of Animals and approved by the Rockefeller University Institutional Animal Care and Use Committee (Protocols #13608, #10005 and #09012).

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
