## [Decision Letter]

Thank you for sending your work entitled “Hypothalamic MCH neurons convey the nutrient value of sugar” for consideration at *eLife*. Your article has been favorably evaluated by a Senior editor, a Reviewing editor, and 3 reviewers, one of whom, Seth Blackshaw, has agreed to reveal his identity.

The Reviewing editor and the reviewers discussed their comments before we reached this decision, and the Reviewing editor has assembled the following comments to help you prepare a revised submission.

This is an elegant and important piece of work, and the manuscript is well constructed and written. It convincingly establishes that the activation of MCH neurons is both necessary and sufficient for the post-ingestive hedonic effects of sucrose. Specifically, the authors start with the paradigm that glucose-containing sugars, such as sucrose, are generally preferred to artificial sweeteners and that the molecular underpinnings of that phenomenon are largely represented by post-ingestive striatal dopamine release. The authors uncover what appears to be the backbone of the neurocircuitry that is responsible for that mechanism. Using optogenetic activation as well as genetically targeted ablation in mice, Domingos and colleagues prove convincingly that neurons identified by production of the neuropeptide MCH are responsible for the differential reward value of sucrose versus sucralose. In addition, they show that MCH neurons indeed project to brain areas involved in reward control and that MCH neurons are required for detection of reward value in mice where sweet perception has been deleted by gene disruption of the TRPM5 receptor. The data are convincing and of high quality.

Substantive concerns:

1) The main target of MCH neuron action in regulating sucrose vs sucralose preference are unclear, as all photostimulation in the study was directly applied to MCH neuronal cell bodies in the LH. The data in Figure 3 demonstrate that MCH efferents are found in proximity to TH-expressing cells in the striatum and midbrain, but only in the VTA are MCH axons and TH-positive cell bodies really intermingled. Since it is also known that MCH neurons project directly to orexinergic neurons, which in turn regulate activity of mesolimbic dopaminergic neurons, this raises the possibility that indirect effects of MCH activity may control sugar preference. Have the authors tried applying photostimulation directly to the VTA, and does it produce behavioral effects similar to those seen in LH? Alternatively, have they shown that MCH neuron activation directly stimulates TH-positive neurons in the VTA or other regions in the central reward circuit?

2) In previous work (7), the authors convincingly showed that leptin negatively regulates sucrose vs sucralose preference, while fasting (presumably coupled with reduction in leptin signaling) enhanced this preference. A key unresolved question here is whether these effects of fasting and leptin on sucrose vs sucralose preference require the action of MCH neurons. It would be highly desirable if the investigators addressed this question using the MCH-Cre;R26-lsl-DTR animals described here.

3) The study does not address whether sucrose vs sucralose preference requires MCH itself. The authors allude to this possibility, noting that *Pmch*^*-/-*^ mice show lower body weight, but this should be easy to test directly, using either genetic or pharmacological approaches. This also addresses the question of whether it is activity of MCH neurons, rather than simply the presence of these cells, that is necessary for sucrose vs sucralose preference. Also, does fructose contribute anything to the “metabolic signal” or is its action simply to increase palatability above that of glucose?

---

## [Author Response]

*This is an elegant and important piece of work and the manuscript is well constructed and written. It convincingly establishes that the activation of MCH neurons is both necessary and sufficient for the post-ingestive hedonic effects of sucrose. Specifically, the authors start with the paradigm that glucose-containing sugars, such as sucrose, are generally preferred to artificial sweeteners and that the molecular underpinnings of that phenomenon are largely represented by post-ingestive striatal dopamine release. The authors uncover what appears to be the backbone of the neurocircuitry that is responsible for that mechanism. Using optogenetic activation as well as genetically targeted ablation in mice, Domingos and colleagues prove convincingly that neurons identified by production of the neuropeptide MCH are responsible for the differential reward value of sucrose versus sucralose. In addition, they show that MCH neurons indeed project to brain areas involved in reward control and that MCH neurons are required for detection of reward value in mice where sweet perception has been deleted by gene disruption of the TRPM5 receptor. The data are convincing and of high quality*.

We greatly appreciate the editors’ and reviewers’ enthusiasm for our work as well as Dr Blackshaw’s willingness to identify himself. Dr. Blackshaw suggests a series of rational and interesting experiments, but in several instances we believe that they may be outside the scope of this paper. Indeed, as outlined below, the studies that Dr Blackshaw suggests are all part of ongoing efforts to further dissect the neural circuit regulating sucrose and reward.

*1) The main target of MCH neuron action in regulating sucrose vs sucralose preference are unclear, as all photostimulation in the study was directly applied to MCH neuronal cell bodies in the LH. The data in*
Figure 3
*demonstrate that MCH efferents are found in proximity to TH-expressing cells in the striatum and midbrain, but only in the VTA are MCH axons and TH-positive cell bodies really intermingled. Since it is also known that MCH neurons project directly to orexinergic neurons, which in turn regulate activity of mesolimbic dopaminergic neurons, this raises the possibility that indirect effects of MCH activity may control sugar preference. Have the authors tried applying photostimulation directly to the VTA, and does it produce behavioral effects similar to those seen in LH? Alternatively, have they shown that MCH neuron activation directly stimulates TH-positive neurons in the VTA or other regions in the central reward circuit*?

We show in the paper that MCH activation can increase dopamine release in striatum using microdialysis, thus confirming that these neurons directly or indirectly activate of VTA neurons. In response to Dr Blackshaw’s comment, we have also generated new data in Figure 5—figure supplement 3 showing that ablation of MCH neurons results in a decreased level of cFos expression in VTA dopamine neurons in response to sucrose ingestion by taste blind TRPM5 knockout mice. These data add further evidence to our conclusion that MCH neurons regulate the activity of DA neurons in the VTA but do not address whether the ability of MCH neurons to activate these neurons is direct or indirect (Dr Blackshaw correctly points out that this could possibly be mediated via orexin neurons or another neural population). This important point is emphasized in the revised Discussion.

*2) In previous work (*[7]*), the authors convincingly showed that leptin negatively regulates sucrose vs sucralose preference, while fasting (presumably coupled with reduction in leptin signaling) enhanced this preference. A key unresolved question here is whether these effects of fasting and leptin on sucrose vs sucralose preference require the action of MCH neurons. It would be highly desirable if the investigators addressed this question using the MCH-Cre;R26-lsl-DTR animals described here*.

The issue of whether leptin alters reward via effects on MCH neurons, or a different neural population(s), is a very important question. We had initially viewed further studies of the precise neural mechanism by which leptin reduces the reward value of sucrose as the subject of a future and comprehensive paper on this topic as the mechanism may or may not involve effects on MCH neurons. In addition, we will have to adopt an alternative strategy to address this question since both MCH neuronal ablation and leptin treatment diminish the reward value of sucrose. Rather, we are planning to assess whether leptin treatment alters the preference of animals for sucralose plus MCH activation vs sucrose. In the revised Discussion, we emphasize the importance of defining the mechanism by which leptin modulates this circuit and because MCH neurons do not express leptin receptor (16), we also discuss the possible role of other neural populations in LH.

*3) The study does not address whether sucrose vs sucralose preference requires MCH itself. The authors allude to this possibility, noting that Pmch*^*-/-*^
*mice show lower body weight, but this should be easy to test directly, using either genetic or pharmacological approaches. This also addresses the question of whether it is activity of MCH neurons, rather than simply the presence of these cells, that is necessary for sucrose vs sucralose preference. Also, does fructose contribute anything to the “metabolic signal” or is its action simply to increase palatability above that of glucose*?

We agree that studies of MCH knockout mice will be important as are analyses of mutations of other neurotransmitters expressed in MCH neurons. This point, as highlighted in the revised Discussion, is also the subject of ongoing studies. We used sucrose in our studies because previous papers on this topic used this nutrient and we did not think that repeating these studies using either fructose or glucose alone would influence our conclusions with respect to our demonstration that MCH neurons are critical for sensing the nutrient value of sucrose. Thus it was our view that studies of either monosaccharide alone were outside the scope of this paper as including these data would essentially require that we repeat all of the studies reported in our paper with both of these sugars.